# RgCop-A regularized copula based method for gene selection in single-cell RNA-seq data

**Snehalika Lall**[1], **Sumanta Ray**[2]\*, **Sanghamitra Bandyopadhyay**[1]\*

**1** Machine Intelligence Unit, Indian Statistical Institute, Kolkata, India, **2** Department of Computer Science and Engineering, Aliah University, Kolkata, India

\* sumanta.ray@aliah.ac.in (SR); sanghami@isical.ac.in (SB)

## Abstract

Gene selection in unannotated large single cell RNA sequencing (scRNA-seq) data is important and crucial step in the preliminary step of downstream analysis. The existing approaches are primarily based on high variation (highly variable genes) or significant high expression (highly expressed genes) failed to provide stable and predictive feature set due to technical noise present in the data. Here, we propose *RgCop*, a novel **reg**ularized **cop**ula based method for gene selection from large single cell RNA-seq data. *RgCop* utilizes copula correlation (*Ccor*), a robust equitable dependence measure that captures multivariate dependency among a set of genes in single cell expression data. We formulate an objective function by adding $l_1$ regularization term with *Ccor* to penalizes the redundant co-efficient of features/genes, resulting non-redundant effective features/genes set. Results show a significant improvement in the clustering/classification performance of real life scRNA-seq data over the other state-of-the-art. *RgCop* performs extremely well in capturing dependence among the features of noisy data due to the scale invariant property of copula, thereby improving the stability of the method. Moreover, the differentially expressed (DE) genes identified from the clusters of scRNA-seq data are found to provide an accurate annotation of cells. Finally, the features/genes obtained from *RgCop* is able to annotate the unknown cells with high accuracy.

## Author Summary

The existing approaches for gene selection which are based on high variation (highly variable genes) or significant high expression (highly expressed genes), failed to provide a stable and predictive feature/gene set. Since single cell data is susceptible to technical noise, the quality of genes selected prior to clustering is of crucial importance in the preliminary steps of downstream analysis. Here, we propose a novel **reg**ularized **cop**ula based method for gene selection that leverage copula correlation (*Ccor*) measure for capturing cell-to-cell variability within the data. The proposed objective function uses an $l_1$ regularization term to penalizes the redundant co-efficient of features/genes. We got significant improvement in the clustering/classification performance of cells over the other state-of-the-art. Due to the scale-invariant property of copula RgCop is impervious to technical noise, an acute issue associated with scRNA-seq data analysis. Moreover, the selected features/genes can

**Funding:** The authors received no specific funding for this work.

**Competing interests:** The authors have declared that no competing interests exist.

be able to determine the unknown cells with high accuracy. Finally, RgCop can be applicable for identifying rare cell clusters or minor subpopulations within the single cell data.

This is a *PLOS Computational Biology* Methods paper.

## Introduction

With the advancement of single cell RNA-seq (scRNA-seq) technology a wealth of data has been generated allowing researchers to measure and quantify RNA levels on large scales [1]. This is important to get valuable insights regarding the complex properties of cell type, which is required for understanding the cell development and disease. A key goal of single cell RNA-seq analysis is to annotate cells within the cell clusters as efficiently as possible. To do this, the basic goal would be to select a few informative genes that can lead to a pure and homogeneous clustering of cells [2, 3]. The task of selecting effective genes among all gene panel that can precisely discriminate cell type labels can be regarded as a combinatorially hard problem.

The usual approach for annotating cells is to cluster them into different groups which are further annotated to determine the identity of cells [4, 5]. This is considered a popular and unsupervised way of annotating different types of cells present in a large population of scRNA-seq data [6–8]. The general pipeline of downstream analysis of scRNA-seq data typically goes through several steps. Starting from the processing of the raw count matrix, the scRNA-seq data is going through the following steps: i) normalization (and quality control) of the count matrix ii) feature selection, and iii) dimensionality reduction [2, 9]. The first step is necessary to adjust discrepancies between samples of individual cells. Several quality measures are also applied to reduces the skewness of the data. The next step identifies the most relevant features/genes from the normalized data. The relevant genes are either selected by identifying the variation (highly variable genes) [10] or can be selected by calculating the expression levels across all cells which are higher than the average value (highly expressed genes).

The selection of top genes has a good impact on the cell clustering process in the later stage of downstream analysis [2, 11]. A good clustering can be ensured by the following characteristics of features/genes [12]: the features/genes should have useful information about the biology of the system, while not including features containing any random noise. Thereby the selected features/genes reduce the data size while preserving the useful biological structure, reducing the computational cost of later steps.

The usual approach of gene selection is based on the high variability of the gene expression label of scRNA-seq data. This process is simple and suffers from several disadvantages: i) as the expression variability is dependent on pseudo-count, it can introduce biases in the data, ii) Next, PCA is applied in downstream analysis for dimensionality reduction which is not suitable for sparse and skewed scRNA-seq data.

In this paper, we present a method for finding the most informative features/genes from large scRNA-seq datasets based on a robust and equitable dependence measure called copula correlation (Ccor). Although the major applications of copula can be found in the domain of time series, finance, and economics, but it is now ripe for application in different domain of bioinformatics such as for modeling directional dependence of genes [13], finding differentially co-expressed genes [14] high dimensional clustering for identification of sub-populations of patients selection [15] and many more. However, the application of copula in single cell domain, particularly on gene selection is less explored.

In this paper, we show that employing a simple $l_1$ regularization term with the proposed objective function, will improve the performance of any clustering/classification model significantly. The objective function is utilized a new robust-equitable copula correlation (Ccor) measure on one hand and a regularization term to control the coefficient on other hand. Thus it is robust due to regularization and not susceptible to noise due to scale-invariant property copula. *RgCop* has major advantages both in clustering/classifying unknown samples and in the identification of meaningful marker detection. The latter point is addressed because novel marker genes for different cell types are identified with the cell clusters. Biologically meaningful marker selection is usually an important step in the downstream analysis of scRNA-seq analysis. This depends on the homogeneity of the cell clusters identified after the gene selection stage. Our proposed method selects the most informative genes that ultimately leads to a homogeneous grouping of cells of the large scRNA-seq data.

Beyond selecting a good informative gene set that leads good clustering/classification of cells, we also demonstrate that our method performs well in completely independent data of the same tissue. We demonstrate this by evaluating the performance of the selected features in completely unknown test samples. We observed that the selected features are equally effective for clustering/classifying the unknown test samples. We further carry out a simulation study on synthetic generated single cell data using splatter to establish the effectiveness of the proposed method. The results show that the proposed method not only select genes with high accuracy, but is also robust and less susceptible to noise.

**Summary of contributions.** The main contributions of the paper are summarized below:

- We provide first regularized copula correlation (Ccor) based robust gene selection approaches from large scRNA-seq expression data. This robustness is a characteristic of our proposed objective function. The method also works equally well for the small sample and large feature scRNA-seq data.

- We derive a new objective function using the copula correlation and regularization term, and theoretically prove that the selected feature set is optimal with respect to the minimum redundancy criterion.

- The objective function of *RgCop* is designed to simultaneously maximize the relevancy criterion and minimize the redundancy criterion among the two sets of features/genes. The regularization term is also added with the objective function so as to control the large coefficient of the relevancy term.

- *RgCop* is able to effectively cluster/classify unseen scRNA-seq expression data. Annotating unknown cells is crucial and is the final goal of scRNA-seq data analysis. We demonstrate the applicability of our framework in this case. We demonstrate that the selected features are effective for clustering completely unseen test data. The annotation of cells can also be done in a supervised way if one can train a classification model with the selected features.

- Our method is less sensitive (robust) against the noises present in the scRNA-seq data. The objective function uses copula-correlation, a robust-equitable measure which has the advantage of capturing the multivariate dependency between two sets of random variables.

## Results

### Workflow

Fig 1 provides a workflow of the whole analysis performed here. Following subsections discussed the important steps:

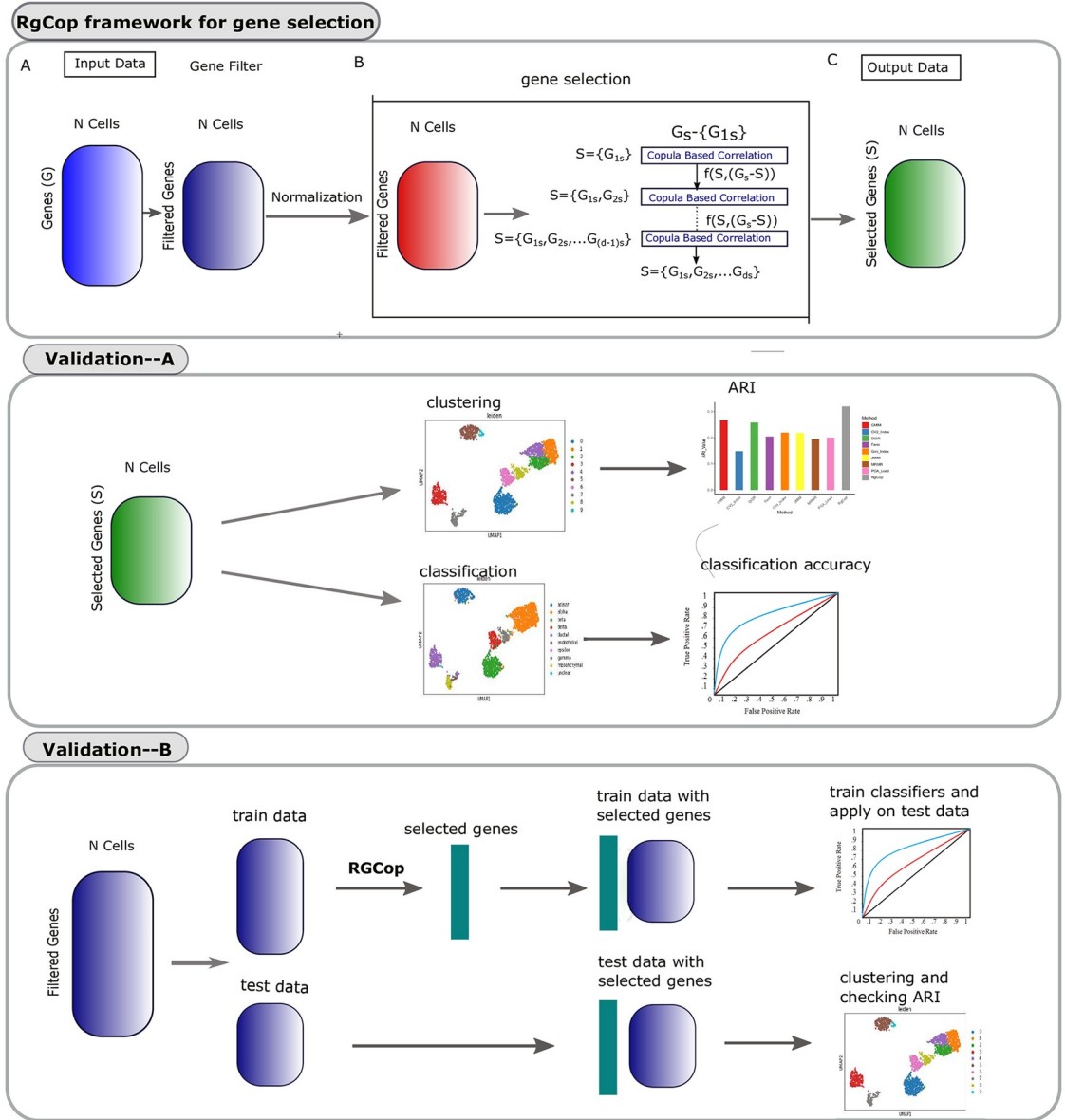

**Fig 1. The whole workflow of the methodology.** *RgCop* framework for gene selection is provided in the top panel. Clustering and classification is performed with the genes obtained from *RgCop* to validate the method (shown in the middle panel). *RgCop* is validated for detection of unknown sample by splitting the data into train-test ratio of 8:2 (shown in the bottom panel). The test data is utilized for validation of the selected genes by *RgCop*.

**A. Preprocessing of raw datasets.** See -A of panel-'*RgCop* framework for gene selection' of Fig 1. Raw scRNA-seq datasets are obtained from public data sources. The counts matrix $M \in \mathcal{R}^{c \times g}$, where $c$ is number of cells and $g$ represents the number of genes, is normalized using a transformation method (Linnorm) [16]. We choose that cells which have more than a thousand genes expression values (non zero values) and choose that genes which have the minimum read count greater than 5 in at least 10% among all the cells. $log_2$ normalization is employed on the transformed matrix by adding one as a pseudo count.

**B. RgCop framework for feature selection.** See -B of panel-'*RgCop* framework for gene selection' of Fig 1. The preprocessed data is used in the proposed copula-correlation (*Ccor*)

based feature/gene selection models. First, a feature ranking is performed based on the *Ccor* scores between all features and class labels. We assume the feature having a larger *Ccor* value is the most relevant one and we include it in the selected list. Next, *Ccor* is computed between the selected relevant features and the remaining features. The feature with a minimum score is called the most essential (and not redundant) feature and included in the selected list. The process continued in an iterative way by including the most relevant and minimum redundant features in each step every time in the list. Feature selection in this way ensures the list of genes will be optimal (see proof of correctness). An $l_1$ regularization term is added with the objective function to penalize the large coefficient of relevancy term. The resulting matrix with selected features is utilized for further downstream analysis.

**C. Validation through clustering.**   See panel-'validation-A' of the Fig 1. We adopt the conventional clustering steps of scanpy [17] package to cluster the resulting matrix obtained from the previous step. We employed two clustering techniques (SC3 [4], and Leiden clustering [18]) for clustering the neighborhood graph of cells. To validate the clusters we utilize the Adjusted Rand Index (ARI) metric which is usually used as a measure of agreement between two partitions. We compare the ARI score of *RgCop* with different state-of-the-art unsupervised feature selection method.

**D. Validation through classification.**   See panel-'validation-A' of the Fig 1. We validate the selected features by employing several classifiers to train the resulting matrix obtained from step-B of the *RgCop* workflow. The features are selected by four widely used supervised feature selection algorithm and the classification accuracy are compared with *RgCop* (see the subsection Comparison with State-of-the-art).

**E. Annotating unknown cells.**   See panel-'validation-B' of the Fig 1. The selected genes obtained from *RgCop* is able to cluster/classify cells of unknown type. The filtered and preprocessed data is divided into train-test ratio 8:2 and the train set is utilized to obtain the selected features using *RgCop*. Several classifier models are trained on the train set with the selected feature set and applied to the test set. The test data with selected features are also used for clustering. This provides the validation of our approach to work in practice.

**F. Marker identification.**   We detect highly differentially expressed (DE) genes within each cluster obtained from step-C in the workflow. Here we utilize Wilcoxon Ranksum test to identify DE genes in each cluster. The top five DE genes are chosen from each cluster according to their p-values.

## Performance on synthetic scRNA-seq data

For single-cell clustering the most common challenge is to discriminate samples between major cell types and its sub-types. Samples of similar cell types tend to overlap within one cluster, discriminating of which required sophisticated method that can extract features from overlapped samples. To explore whether RgCop can address this issue we apply it on simulated data generated by a widely used method called Splatter [19]. We make four experimental setup ($S_1$ to $S_4$) to comprehensively evaluate RgCop. Splatter is utilized to generate the data in each case.

$S_1$: generated four groups of 500 cells with the sample ratio of $10 : 10 : 10 : 70$. Low dropout rate is set ($\sim 0.2$) over 2000 genes.

$S_2$: generated four equal-sized groups of cells, each group consisting 25% of the total (500) cells, over 2000 genes at a high dropout rate ($\sim 0.5$).

$S_3$: generated four groups of 500 cells, with the sample ratio $10 : 10 : 10 : 70$ over 2000 genes with a high dropout rate ($\sim 0.5$).

**Table 1. Four setups for generating simulated datasets using Splatter [19].**

| Setups | Group Proportions (%) | Dropout rate | DE Gene Proportion (%) |
|--------|----------------------|--------------|------------------------|
| S1 | (10, 10, 10, 70) | 0.2 | 40 |
| S2 | (25, 25, 25, 25) | 0.5 | 40 |
| S3 | (10, 10, 10, 70) | 0.5 | 10 |
| S4 | (25, 25, 25, 25) | 0.2 | 10 |

$S_4$: generated four equal-sized groups of 500 cells over 2000 genes at a low dropout rate $\sim 0.2$.

The proportions of differentially expressed (DE) genes in $S_1$ to $S_4$ were selected as 40%, 40%, 10%, 10% respectively. The details of the simulation settings are shown in Table 1.

To tune the regularization parameter $\gamma$ (see Eq 6), the feature selection process is repeated for nine set of values ranging from 0 to 0.5 ($\gamma = \{0, 0.002, 0.005, 0.009, 0.02, 0.07, 0.09, 0.3, 0.5\}$, see Method section for explanation). We trained random forests classifier to measure overall accuracy over 100 simulation replicates. Table 2 reports median accuracy for the nine $\gamma$-values in four simulation setups. High accuracy is observed for the $\gamma$-parameter in the range of $\gamma \in [0.07, 0.3]$ (see Table 2). The selected range of $\gamma$ values are utilized for the later stage of analysis.

## Comparison with state-of-the-art

We compared the efficacy of *RgCop* by comparing with four well known techniques for identifying highly dispersed genes in scRNA-seq data: *Gini Clust* [20], *PCA Loading* [21], *CV² Index* and *Fano Factor* [22]. We also compared the performance of *RgCop* with four widely used supervised feature selection techniques: CMIM [23], JMIM [24], DISR [25], MRMR [26]. A short description of competing methods and parameter settings is described in the S1 Text, sec-1.

**Clustering performance on real dataset using unsupervised method.** Here single cell Consensus clustering (SC3) method [4] is employed for clustering expression matrix with selected features. Fig 2, panel-A illustrates the boxplots of ARI Values of the clustering results on Yan, Muraro, and Pollen datasets. We vary the number of selected features in the range from 500 to 1000 and compute the ARI scores for each method. It can be seen from the figure that *RgCop* achieves high ARI values in almost all the three datasets. For the Yan dataset, while the performance of other methods is relatively low, *RgCop* achieves a good ARI value, demonstrating the capability of *RgCop* to perform in small sample data. We also create a visualization of the clustering performance of *RgCop* in Muraro, yan, and Pollen datasets. Fig 2B, shows two dimensional t-SNE plot of predicted clusters and their original labels. Fig 2C shows heatmaps of cell × cell consensus matrix representing how often a pair of cells is assigned to the same cluster considering the average of clustering results from all combinations of clustering

**Table 2. Classification accuracy are reported for different values of $\gamma$ using RgCop.**

| Method | Classifier | Setups | q-Values | | | | | | | | |
|--------|-----------|--------|-----|-------|-------|-------|------|------|------|------|------|
| | | | 0 | 0.002 | 0.005 | 0.009 | 0.02 | 0.07 | 0.09 | 0.3 | 0.5 |
| RgCop | Random Forest | S1 | 0.85 | 0.90 | 0.90 | 0.90 | 0.91 | 0.93 | 0.94 | **0.95** | **0.95** |
| | | S3 | 0.78 | 0.81 | 0.80 | 0.81 | 0.81 | 0.84 | **0.87** | **0.90** | **0.89** |
| | | S2 | 0.96 | 0.98 | 0.99 | **0.99** | **0.99** | **0.99** | **0.99** | **0.99** | **0.98** |
| | | S4 | 0.86 | 0.90 | 0.91 | 0.91 | 0.90 | 0.90 | 0.90 | **0.98** | **0.97** |

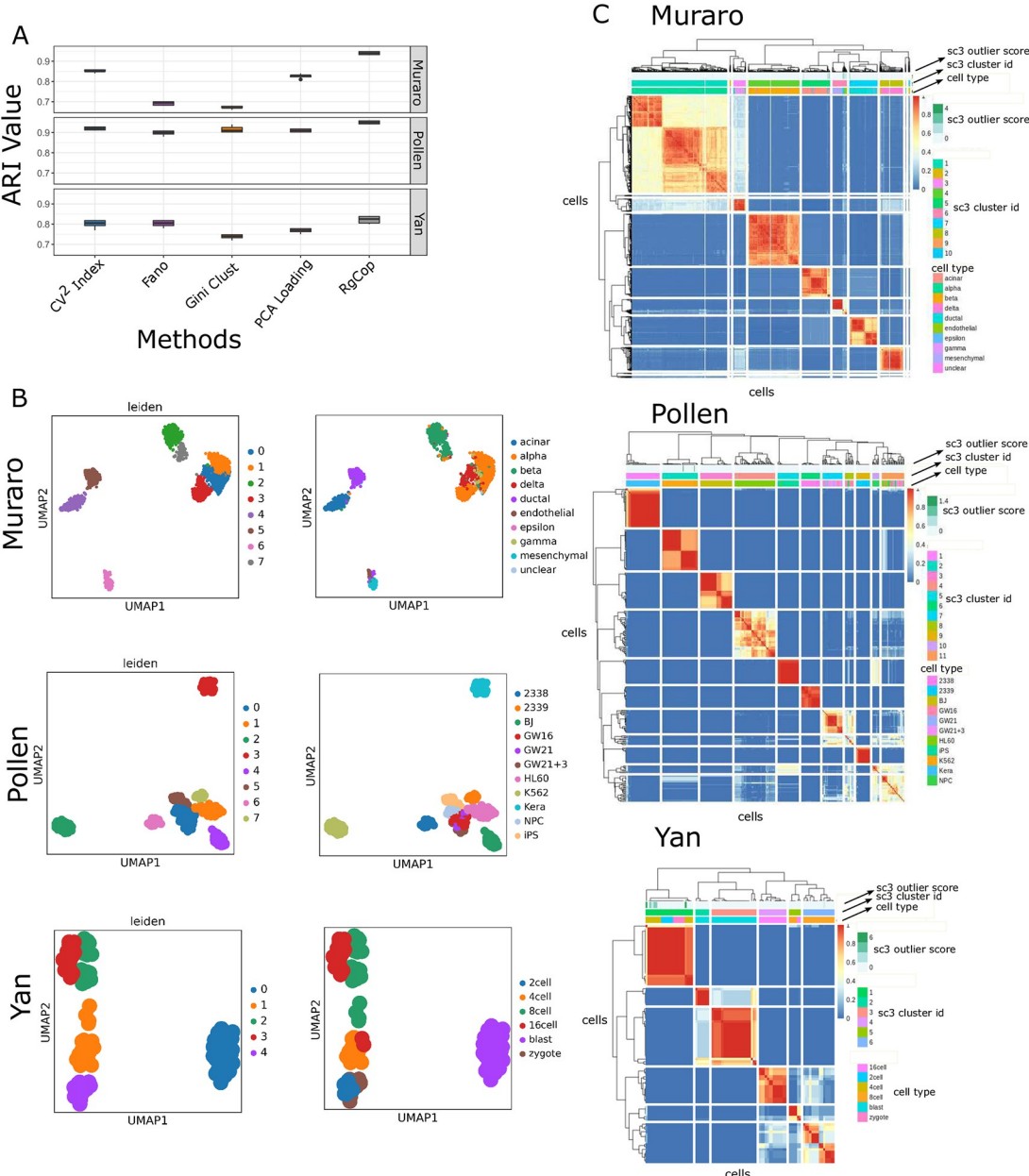

**Fig 2. Figure shows the comparisons of clustering performance.** Panel-A shows the boxplot of ARI values computed from the clustering results of each competing method. Each box represents ten ARI scores of clustering results for selected 6 sets of features ranging from 500 to 1000. Panel-B shows the 2 dimensional UMAP visualization of clustering results of three datasets for *RgCop*. Panel-C shows the consensus clustering plots of obtained clusters from *RgCop*.

parameter [4]. Zero score (blue) means two cells are always assigned to different clusters, while score '1' (red) represents two cells are always within the same cluster. The clustering will be perfect if all diagonal blocks are completely red and off-diagonals are blue. A perfect match between the predicted clusters and the original labels can be seen from panel-B and panel-C of Fig 2.

**Classification performance on real dataset using supervised method.** We compare *RgCop* with four well known supervised feature selection methods and compute the

**Table 3. Classification results on real datasets using supervised methods.**

| Classifier | Dataset Name | MRMR | DISR | JMIM | CMIM | *RgCop* ($\gamma = 0.3$) |
|---|---|---|---|---|---|---|
| GBM | Muraro | 0.81 ± 0.05 | 0.79 ± 0.04 | 0.78 ± 0.010 | 0.8 ± 0.02 | **0.88 ± 0.009** |
| | Pollen | 0.87 ± 0.03 | 0.82 ± 0.01 | 0.8 ± 0.05 | 0.82 ± 0.02 | **0.94 ± 0.002** |
| | Yan | 0.97 ± 0 | 0.96 ± 0.02 | 0.97 ± 0.02 | 0.96 ± 0 | **0.98 ± 0.01** |
| NNET | Muraro | 0.72 ± 0.02 | 0.71 ± 0.01 | 0.71 ± 0.05 | 0.74 ± 0.03 | **0.79 ± 0.07** |
| | Pollen | 0.81 ± 0.01 | 0.82 ± 0.003 | 0.89 ± 0.02 | 0.80 ± 0.01 | **0.91 ± 0.02** |
| | Yan | 0.99 ± 0.01 | 0.98 ± 0.02 | 0.98 ± 0.03 | 0.98 ± 0.003 | **0.99 ± 0.02** |
| SVM | Muraro | 0.77 ± 0.04 | 0.78 ± 0.01 | 0.77 ± 0.011 | 0.79 ± 0.01 | **0.85 ± 0.01** |
| | Pollen | 0.84 ± 0.06 | 0.85 ± 0.03 | 0.80 ± 0.03 | 0.81 ± 0.01 | **0.93 ± 0.001** |
| | Yan | 0.98 ± 0.03 | 0.98 ± 0.02 | 0.96 ± 0.01 | 0.98 ± 0.04 | **0.98 ± 0.003** |

classification accuracy. Three widely used classifiers are considered in our work, Neural Network (*NNET*), Support Vector Machine (*SVM*), and Gradient Boosting Machine (*GBM*) for learning the expression matrix with selected features. Table 3 shows the average test accuracy and the corresponding standard errors over 50 runs for each of the competing methods.

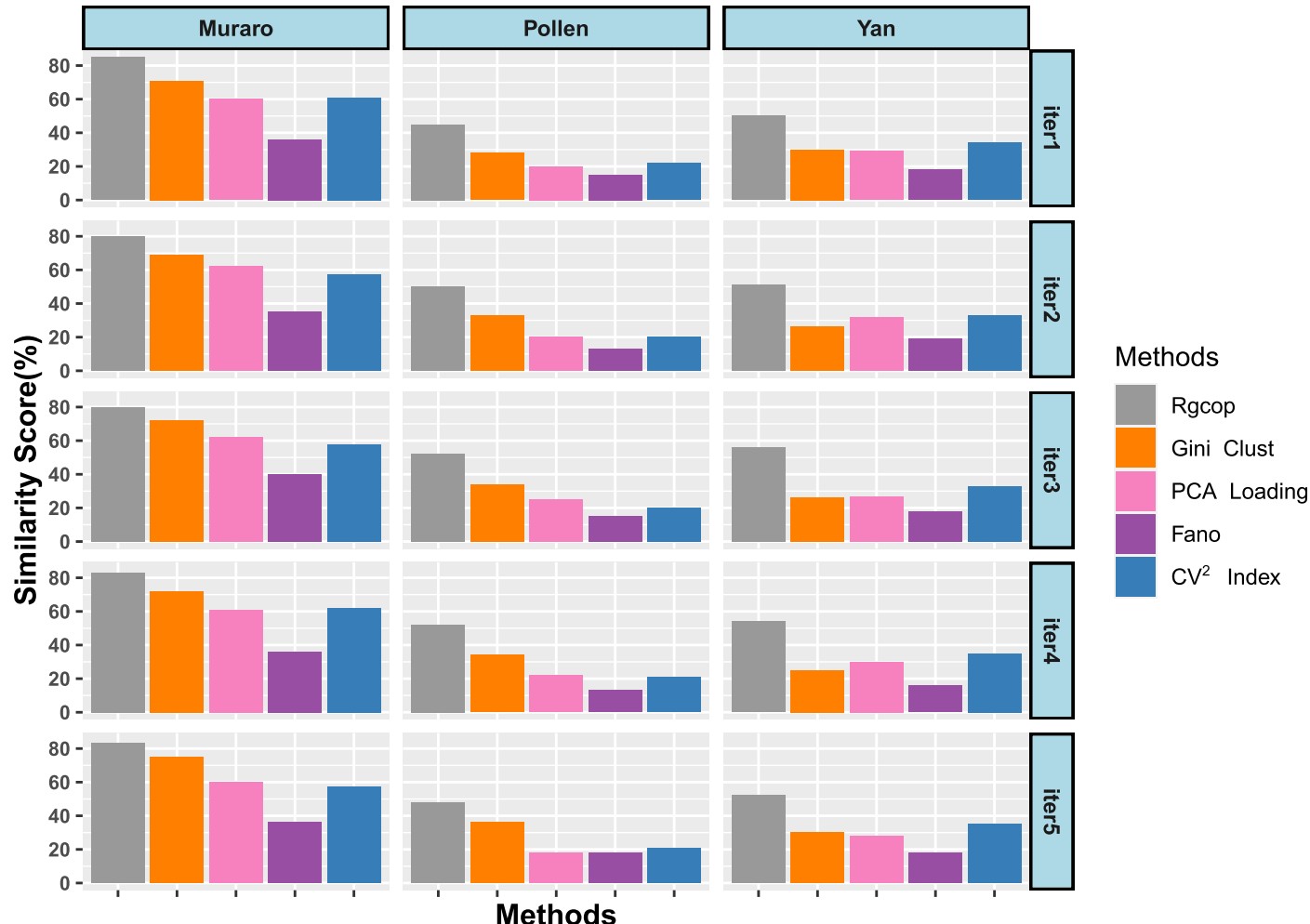

**Fig 3. Figure shows the median of *SimilarityScore* (percentage) of five different competing methods.** Five iterations (iter1,iter2, iter3, iter4, iter5) are performed with 100 repetition in each iteration to compute the median of *SimilarityScore*.

Results demonstrate that RgCop outperforms the other existing supervised feature selection methods.

## Stability performance

Protocols for preprocessing of scRNA-seq data are complex and often suffer from technical biases that vary across cells. This causes biases in the downstream analysis if the noise is not properly handled. *RgCop* utilizes copula which is well known for its scale invariance property that makes it robust against noise in the data. To show the performance of *RgCop* in noisy data, white Gaussian noise with a mean ($\mu = 0$) and standard deviation 1 is mixed to each gene/feature of a dataset. The function *Add.Gaussian.noise* of R package RMThreshold is used to generate Gaussian noise. Next, relevant 500 genes/features are chosen from each of the noisy datasets, and the percentage of matching is computed with the original genes/feature sets. We define a matching feature score (percentage) as follows $SimilarityScore = ((N - D)/N) * 100$, where $N$ represents the total number of features, and $D$ represents the number of feature discrepancies between the original and noisy dataset. We perform 5 iterations, each contains 100 such experiments. For a competing method, each trial gives 100 scores for one dataset and the median of these scores are shown in Fig 3. Each row of the figure shows bar plots of the median values for three scRNA-seq datasets. It can be observed from the figure that *RgCop* achieves better *SimilarityScore* for all the datasets, particularly for small sample data.

## Classifying test samples using the selected features

Classifying new cell samples is crucial for the scRNA-seq data analysis pipeline. Here, we address this by performing an analysis to show how the selected genes are important for discriminating the unknown cell samples. We first split the data in train-test ratio of 8:2 and use RgCop to select 500 most informative genes from the training set. Next, we train a random forest classifier with this data and retain the trained model. Table 4 shows the classification performance of the trained model on the test sample using the selected genes as the feature set. The experiment is repeated 100 times with a random split of train-test data with 8:2 ratio in each case. High classification accuracy demonstrates that the selected feature sets are equally important for discriminating the cells of the completely independent test samples.

## Marker gene selection

We have chosen marker genes (DE genes) for different cell types from the clustering results. Differentially Expressed (DE) genes are identified from every cluster using Wilcoxon rank-sum test. We use this to directly assess the separation between distributions of expression profiles of genes from different clusters. Fig 4 illustrates the top five DE genes from each cluster of Pollen dataset (panel-A), and Yan dataset (panel-B). The higher expression values of the top five DE genes (shown in the heatmap of panel-A, B) for a particular cluster suggests the

**Table 4. Classification accuracy on test datasets using RgCop.**

| Datasets | Classifier | RgCop |
|----------|------------|-------|
| Yan | Random Forest | $0.94 \pm 0.05$ |
| Muraro | | $0.88 \pm 0.01$ |
| Pollen | | $0.97 \pm 0.01$ |
| PBMC | | $0.67 \pm 0.05$ |

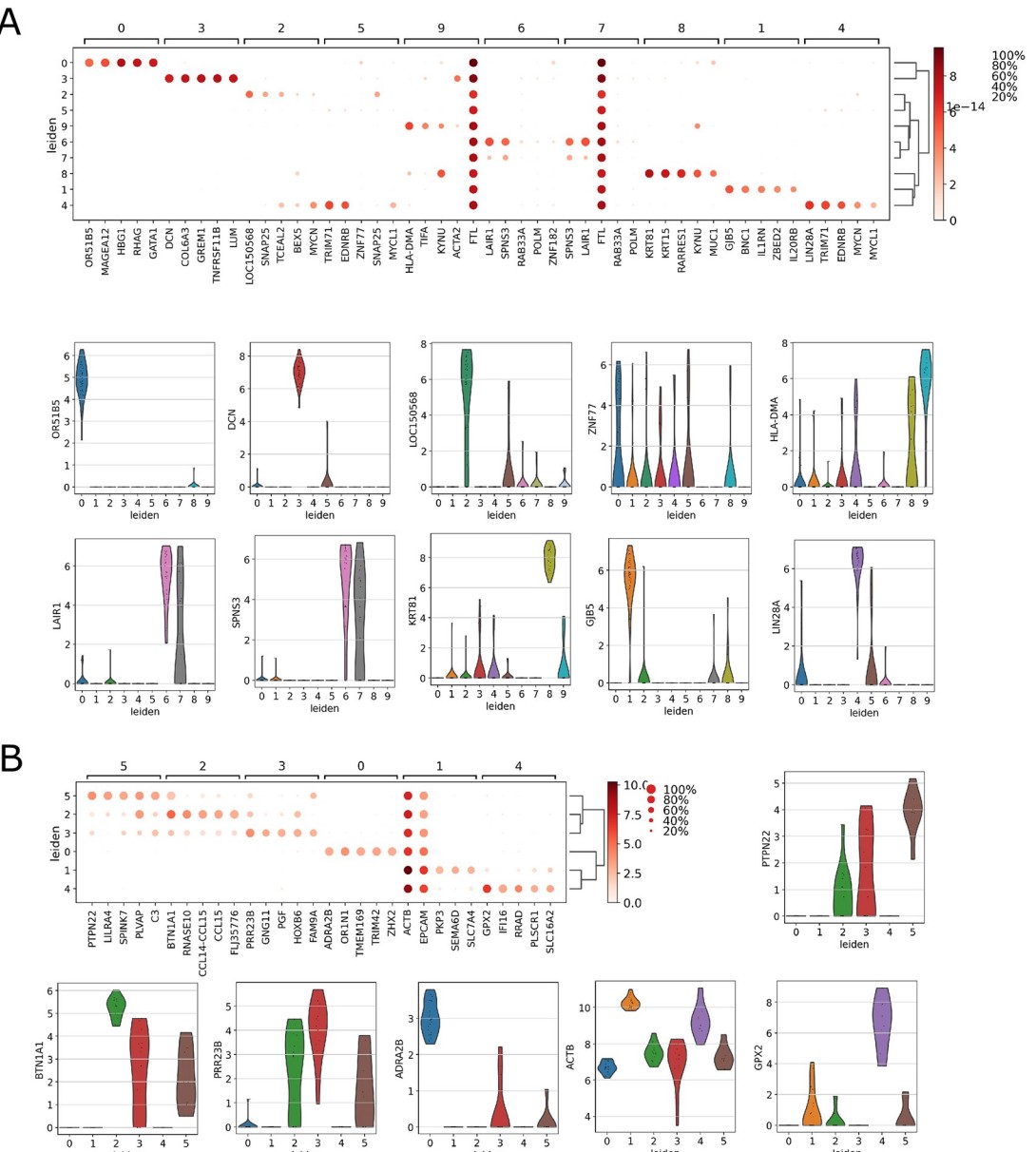

**Fig 4. Figure shows marker analysis for Pollen dataset (panel-A), and Yan dataset (panel-B).** The average expression values of the top five DE genes are shown in heatmap of panel-A, and -B. The violin plots of the expression profiles of those top DE genes within each cluster are shown in panel-A and -B.

presence of marker genes within the selected gene sets. The results are detectable from violin plots of the expression profiles of top DE genes within each cluster (Fig 4A and 4B).

To know how the identified DE genes can be utilized to annotate the cell cluster we performed an analysis. We matched the cluster specific DE genes of PBMC data with experimentally verified cell markers downloaded from CellMarker [27] and annotate the cell clusters with a specific type. Table 5 shows the overlaps between identified DE genes with markers of a specific type. We utilized these genes to annotate the cell clusters and compute the accuracy of annotation. As the original labels are known, so we can verify the annotations with the original

**Table 5. Overlaps of cluster specific DE genes with markers of specific cell type.**

| Dataset | cell type | markers (pubmed id) |
|---------|-----------|---------------------|
| PBMC | Regulatory T cell | IL32 (30093597) |
| | CD8 T cell | CCL5 (30093597) |
| | NK cells | NKG7 (8458737), GNLY (12884856) |
| | Effector CD8+ memory T cell | GZMH (28622514) |
| | Plasmacytoid dendritic cells | GZMB (19965634) |
| | CD4+ cytotoxic T cell | CST7 (28622514) |
| | B cell | CD79A (11396639), CD37 (24952935) |
| | Monocyte derived dendritic cells | CST3 (19956698) |
| | Megakaryocyte progenitors | PPBP (27084257), PF4 (30645026) |

cell labels. High accuracy (acc = 0.78) reveals a good and approximately accurate annotation of cell clusters of PBMC data.

## Rare cell identification

We designed an experiment to evaluate the performance of RgCop to detect rare cellular identities. For this we used two scRNA-seq datasets, the first one is comprising 293T and Jurkat cells mixed in vitro [28] and the second one is PBMC68K. After applying RgCop, the selected features are used for clustering the datasets. For PBMC68k we utilized a graph based clustering technique ('Leiden' Algorithm) which is frequently used in most of the standard pipelines for scRNA-seq data analysis like Seurat and scanpy. For the simulated data consisting of JurKat and 293T cells we used a simple k-means algorithm with $k = 2$. We observed that the identified genes have the ability to discover CD14+ monocytes (4.8% of the total cells) and Dendritic subtypes (2.7% of the total cells) from PBMC68k (Fig 5A and 5B) and Jurkat cells (2.5% of the total cells) (Fig 5C and 5D) from the simulated data. The higher adjusted rand index for the PBMC68k clustering (see Fig 5E) also supports the efficacy of RgCop to use in single-cell clustering.

## RgCop is robust for data with different batches

To know the efficacy of RgCop in data of different batches we performed this analysis. We downloaded two processed datasets from the github repository (https://github.com/ JinmiaoChenLab/Batch-effect-removal-benchmarking) of Tran et al. [29]. The first one consists of human blood dendritic cell (DC) cells analyzed with Smart-Seq2 technology in two batches. Both of the batches share 96 pDC and 96 double negative cells. Each batch has one biologically similar unshared cell type: CD141 cells and CD1C cells present in the first batch and second batch respectively. Both batches consist of 288 cells and 16,594 genes. We applied RgCop in the first/second batch and cluster the second/first batch with the selected genes. In both cases, we got high ARI scores (0.819 and 0.877 for two cases respectively). The CD141 cells and CD1C cells are correctly discovered in respective batch using the selected genes. For each case, we select 500 genes for clustering.

The second data is composed of three batches: 2885 cells of 293T cells are present in batch-1, 3258 jurkat cells are in batch-2, and batch-3 consists of a 50/50 mixture of Jurkat and 293T cells. The gene expression data contains 16,602 genes obtained using the 10x Genomics platform. We select genes from batch-3 data using RgCop and cluster batch-1 and batch-2 separately. We got an ARI score of 0.891 and 0.816 for batch-1 and batch-2 clustering respectively.

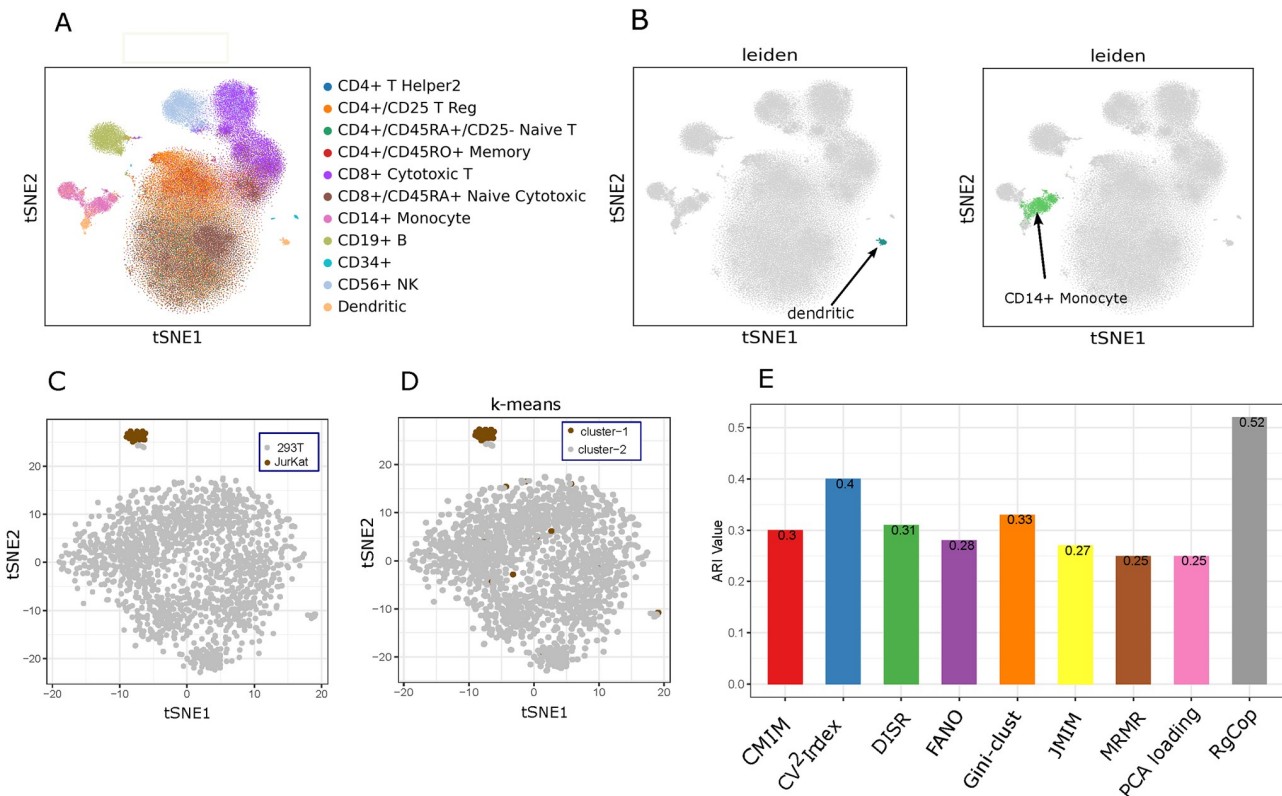

**Fig 5. Results of RgCop applied in PBMC68k and a simulated dataset consisting of JurKat and 293T cells.** A and C: The tSNE based 2-dimensional embedding of PBMC68K and the simulated data. B and D: Rare clusters identified using the features selected by RgCop. E: Comparisons of ARI scores among all the competing methods (all supervised and unsupervised methods) in PBMC68K data.

## A case study on ultra large scRNA-seq data

*RgCop* is applied on large single cell dataset consisting of 690k cells of Adult Mouse Brain [30]. We downloaded the Digital Gene Expression (DGE) matrices of cells from http://dropviz.org/. Cells were first assigned to clusters and sub-clusters according to the *cell_cluster_outcomes. RDS* files. Annotation of cells are incorporated based on the assigned clusters and sub-clusters downloaded from https://storage.googleapis.com/dropviz-downloads/static/annotation. BrainCellAtlas_Saunders_version_2018.04.01.RDS. We pre-processed the data using the standard pipeline used by Seurat. After preprocessing the data, *RgCop* selects 500 genes which are utilized for clustering. Leiden graph-clustering method [18] are utilized to cluster the cells with selected genes (see Fig 6B). Among the 19 cell types (see Fig 5A), RgCop can capture the clusters with minor cells e.g. endothelial tip/stalk, polydendrocyte-1/2, mural (Fig 6C). Moreover, the clustering achieves higher ARI scores (ARI = 0.78),which shows the efficacy of RgCop for selecting the discriminative features from ultra-large single cell data.

## Execution time

All experiments were carried out on a Linux server having 50 cores and *X86_64* platform. As our proposed method is a wrapper-based step wise feature selection method, so it takes more time than any filter-based feature selection technique (e.g. $CV^2$ *index*, *Gini−clust*). To check how the competing methods scale with the number of cells (and classes) we performed an analysis. We have generated simulated data (using splatter) by varying the number of cells

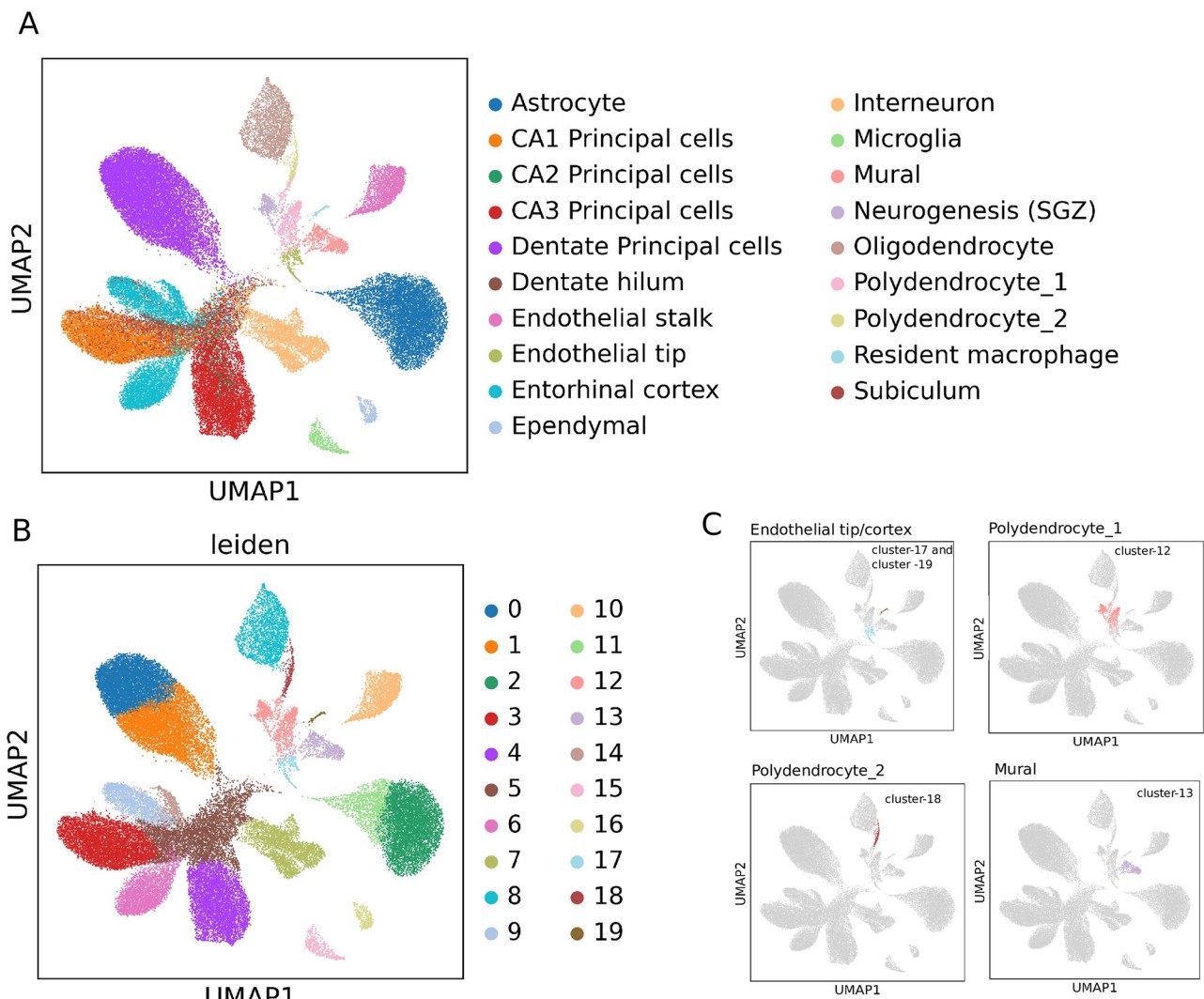

**Fig 6. Results of RgCop applied on adult mouse brain data of Saunders et al. [30].** A. Clustering of cells with original annotation. B. 19 cell clusters are identified by Leiden graph clustering algorithm. Clustering is performed with 500 selected genes of RgCop. C. Some clusters contain minor cell subpopulations such as Endothelial tip/stalk, polydendrocyte-1/2.

(and classes). Four simulated data are generated with the number of cells and classes as follows: 500 cells with two classes, 1000 cells with three classes, 1500 cells with four classes, and 2000 cells with five classes. All data are generated with equal group probabilities, 2000 number of features, fixed dropout rate (0.2), and 40% DE gene proportion. 500 features are selected in each case and the runtime is compared with different competing methods. The execution time (minute) for each dataset is given in Table 6.

## Conclusions

The selection of informative genes in scRNA-seq data is crucial and an essential step for the downstream analysis. Because of the large feature/gene set of scRNA-seq data, selecting important genes is a challenging task which has an immense effect in clustering and annotation results in the later stage of downstream analysis. The proposed method *RgCop* addressed this

**Table 6. Execution time in minute for five competing methods.**

| Datasets | # Selected Feature | # Cells | # Class | Execution Time (in Minute) | | | | |
|---|---|---|---|---|---|---|---|---|
| | | | | RgCop | Gini Clust | $CV^2$ Index | Fano | PCA Loading |
| Data1 | 500 | 500 | 2 | 9 | 2 | 1 | 1 | 3 |
| Data2 | | 1000 | 3 | 13 | 2 | 1 | 1 | 7 |
| Data3 | | 1500 | 4 | 17 | 3 | 1 | 3 | 11 |
| Data4 | | 2000 | 5 | 20 | 5 | 3 | 5 | 14 |

task by employing a robust and equitable dependence measure called copula-correlation (*Ccor*). It can accurately measure relevancy and redundancy simultaneously between two sets of gene. *RgCop* also add simple $l_1$ regularization technique with its objective function to control the large coefficients of relevancy terms. Realistic simulations confirm the utmost accuracy of the *RgCop* in simulated single cell data. We also demonstrated that *RgCop* results high accuracy both in clustering and classification performance with the selected genes from real-life scRNA-seq datasets. The identified marker genes can also be able to dissect cell clusters, suggesting the inclusions of marker genes within the selected sets. *RgCop* also performed well in the datasets which are coming with different batches.

*RgCop* introduces a stable feature/gene selection which is evaluated by applying it in noisy data. By virtue of the important *scale invariant property* of copula, the selected features are invariant under any transformation of data due to the most common technical noise present in the scRNA-seq experiment. The range of tuning parameter (regularization coefficient ($\lambda$)) is determined using *RgCop* on synthetic single cell dataset generated from Splatter [19]. *RgCop* also produces accurate clustering/classification results on four sc-RNA seq datasets. The results are validated using ARI score/classification accuracy. The stability of *RgCop* is evaluated by applying it in noisy data and matching the resulting feature set with the original one. This was performed multiple times with varying number of selected features. The resulting ARI scores utilize a minimum deviation suggesting a robust and stable approach for feature selection.

It can be noted that although *RgCop* primarily detect variable genes from scRNA-seq data, we extended the process by employing a clustering/classification technique with it to annotate unknown cells. The efficacy of *RgCop* is demonstrated by applying it to cluster/group unknown cells with the selected genes/features. A precise annotation of cell clusters also illustrates the applicability of *RgCop* to select the most variable genes in the early stage. The most general classifiers trained with the selected features can accurately predict the cell types of an unknown sample. Several genes are highlighted having a high expression level within clusters, which are acting as markers.

Taken together, the proposed method *RgCop* not only outperforms in informative gene selection but also able to annotate unknown cells/cell-clusters in scRNA-seq data. It can be observed from the results that *RgCop* leads both in the domain of robust gene/feature selection and type annotation of unknown cell in large scRNA-seq. The results prove that *RgCop* may be treated as an important tool for computational biologist to investigate the primary steps of downstream analysis of scRNA-seq data.

## Method

### Datasets description

**Single cell RNA sequence datasets.** The study used the following single-cell RNA sequence datasets: Yan [31], Pollen [32], Muraro [33] and PBMC68k [1] (see Table 7). A detailed description of the data is provided in the following text.

**Table 7. A brief summary of the real scRNA sequence dataset.**

| Dataset Name | Features | Instances | Class |
|---|---|---|---|
| Yan | 20214 | 90 | 7 |
| Muraro | 19127 | 2126 | 10 |
| Pollen | 23794 | 299 | 11 |
| PBMC | 32738 | 68793 | 11 |

- Yan: The dataset consists of a transcriptome of 124 individual cells from a human preim-plantation embryo and embryonic stem cell. The 7 unique cell types accommodates labelled 4-cell, 8-cell, zygote, Late blastocyst, and 16-cell.[GEO under accession no. GSE36552; [31]]. We downloaded the processed data from https://hemberg-lab.github.io/scRNA.seq.datasets/human/edev/ which contains 20214 features and 90 samples.

- Pollen: Single cell RNA seq pair-end 100 reads from single cell cDNA libraries were quality trimmed using Trim Galore with the flags. It contains 11 cell types. [GEO accession no GEO1832359; [32]]

- Muraro: Single-cell transcriptomics was carried out on live cells from a mixture using an automated version of CEL-seq2 on live, FACS sorted cells. It contains 2126 number of cells. It is a human pancreas cell tissue with 10 cell types. The dataset was downloaded from GEO under accession no GSE85241 [33].

- PBMC68k: The dataset [1], is downloaded from 10x genomics website https://support.10xgenomics.com/single-cell-gene expression/datasets. The data is sequenced on Illumina NextSeq 500 high output with 20,000 reads per cell.

## Background theory that supports *RgCop*

**Short description on Copula.** The 'Copula' term [34] is originated from a Latin word *copulare*, which joins multivariate distributions to its one dimensional distribution function. The copula is considerably employed in high dimensional datasets to obtain joint distributions using uniform marginal distributions and vice versa. See S1 Text sec-2 for a detailed descriptions of copula and its related measures.

**Copula correlation measure.** Let, $Y = \{y_1, y_2\}$ and $Z = \{z_1, z_2\}$ are two bivariate random variables and their joint and marginal distributions are $H_{YZ}$, $F_Y(y)$ and, $F_Z(z)$ respectively. Now $H_{YZ}$ can be expressed as: $H_{YZ}(y, z) = C(F_Y(y), F_Z(z))$, where $C$ is a copula function.

Kendall tau($\tau$), the measure of association, [35] can be expressed in terms of concordance and discordance between random variables. Kendall tau is the difference between probability of concordance and discordance of $(y_1, y_2)$ and $(z_1, z_2)$. It can be described as

$$\tau_{YZ} = [P(y_1 - y_2)(z_1 - z_2) \geq 0] - [P(y_1 - y_2)(z_1 - z_2) \leq 0] \tag{1}$$

According to Nelson [36] Kendall tau can be expressed using copula function:

$$\tau(C_{Y,Z}) = \tau_{YZ} = 4 \iint\limits_{0}^{+1} C(u, v) \, dC(u, v) - 1 \tag{2}$$

Where, $u \in F_Y(y)$ and $v \in F_Z(z)$. $\tau(C_{Y,Z})$ is termed as copula-correlation (*Ccor*) in our study.

**A note on regularization.** Regularization is a type of regression that penalizes the coefficient of redundant feature towards zero [37] (see S1 Text sec-3 for detailed description). The simplest regularization is $l_1$ norm or Lasso Regression, which adds "absolute value of magnitude" of coefficient as penalty term to the loss function. For any vector $A \in \mathcal{R}^m$, the $l_1$ norm is $||A||_1 = \gamma \sum_{i=1}^m |A_i|$, where $\gamma$ is a tuning parameter, controls penalization. For $\gamma = 0$ regularization effect is none. When $\gamma$ value increases, it starts to penalizes the larger coefficients to zero. However, after a certain value of $\gamma$, the model starts losing important properties, increasing bias in the model and thus causes under-fitting. We tuned $\gamma$ using simulated single cell datasets generated by the well known algorithm Splatter [19].

## Gene selection using *RgCop* algorithm

**Max relevancy.** A gene $g_i$ is more relevant to class labels $C_D$ than another gene $g_j$, if $g_i$ has higher *Ccor* score with $C_D$ than $g_j$. This is called *Relevancy* test for the genes [38] and is used to select most relevant gene from a gene set. Formally it can be described as: $g_i \prec g_j$ if the following is true,

$$\tau(C_{g_i, C_D}) > \tau(C_{g_j, C_D}), \tag{3}$$

Where $\tau[C(x, y)]$ represents copula correlation between two random variable $X$ and $Y$. For estimating the copula density we have used empirical copula. The maximum-relevancy method choose the gene (feature) subset among gene set $G$ as

$$G_{max-relevancy} = \arg \max_{g_i \in G} \frac{1}{|G|} \sum_{g_i \in G} \tau(C_{g_i, C_D}). \tag{4}$$

$G_{max-relevancy}$ may contains genes that are mutually dependent. This is because we only consider the *Ccor* between a gene and class labels, overlooking the mutual dependency among the selected and non-selected genes. This results spurious genes in the selected list.

**Min redundancy.** Redundancy is a measure that computes the mutual dependence among set random variables. Here we used the same definition of *Ccor* to compute multivariate dependency between selected gene ($g_i$) and non selected gene sets ($g_s$). Formally it can be expressed as

$$G_{min-redundancy} = \arg \min_{s:g_s \in S} \tau(C_{g_i:g_s}) \tag{5}$$

**Objective function.** *RgCop* utilizes a forward selection wrapper approach to select gene iteratively from a gene set. It uses multivariate copula-based dependency instead of the classical information measure. The objective function integrates the relevancy and the redundancy terms defined using the *Ccor*. Mathematically, it can be expressed as follows.

Let us assume genes ($g_1, \ldots, g_i$) are in the selected list $G_s$. The next gene $g_{i+1} \in (G - G_s)$ in at ($i$+1) iteration is using the objective function

$$f = \arg \max_{g_i \in (G-G_s)} [(\tau(C_{g_i:C_D}) - \tau(C_{g_i:g_1:g_2:\cdots:g_s}))$$
$$+ \gamma ||\tau(C_{g_i:C_D}) * Var(g_i)||_1] \tag{6}$$

Where, $\tau(C_{g_i:g_j}) = [4 \iint_0^{+1} C(g_i; g_j) \, dC(g_i; g_j) - 1]$ is Kendall tau dependency score of Empirical copula between two genes $g_i$ and $g_j$. Here, $\gamma$ represents the regularization coefficient. An overview of the *RgCop* algorithm is given in algorithm-1

**Algorithm 1** $l_1$ Regularized Copula Based Feature Selection (*RgCop*)

```
Input: Preprocessed Data Matrix D, Cell Type C_D, Number of Selected
Features, d.
Output: Optimal Feature subset,(G_s).
  Initialisation:
  G_s = ∅, {T will hold sub-set feature indices.}
  g_1s ← argmax_{g_i} τ(C_{g_i, C_D}), {Maximum Relevancy}
  for all i = 0 to (d − 1) do
    E = ∅
    R ← τ(C_{g_i; C_D}), {Relevancy Criterion}
    S ← τ(C_{g_i; g_1s;...; g_ds}), {Redundancy Criterion}
    E ← (R − S)
    G_s ← {G_s ⋃ arg max (lim_{g_i} {E})
    G ← G − {g_i}
  end for
  return G_s
```

**Proof of correctness.** Suppose $G_s = \{g_1, g_2, \ldots, g_i, g_{i+1}, \ldots g_d\}$ denotes a subset of genes obtained from a gene set $G$ using *RgCop*. Here $g_i$ represents selected gene at iteration $i$. We claim that the set $G_s$ is optimal.

**Proof.** Let us prove this by the method of contradiction. If we assume the claim is not true, then there should exist some another optimal gene set $G_s'$. Without loss of generality, let us assume $G_s'$ has a maximum number of initial genes ($i$ number genes) common with $G_s$.

Now $G_s'$ can be written as $G_s' = \{g_1, g_2, \ldots, g_i, g_k, \ldots, g_d\}$. So, $G_s'$ contains $\{g_1, g_2, \ldots, g_i\}$ from $G_s$, but not $g_{i+1}$. Following our assumption $g_{i+1}$ cannot be included in any of the optimal gene lists ($G_s'$ has maximum $i$ number of initial genes overlapped with $G_s$).

Now we claim that $k > (i+1)$. This is because $k$ cannot be equal to $i+1$, otherwise $G_s'$ would have ($i+1$) genes overlapped with $G_s$. Similarly, $k \nleq i$, because otherwise $G_s'$ will contains redundant genes.

Now by the definition of our objective function ($f$) we can write: $f(g_k) < f(g_{i+1})$. So we can substitute $g_k$ with $g_{i+1}$ in the $G_s'$ list, and the list will be still optimal. This contradicts our assumption that $g_{i+1}$ cannot be included in any optimal list. This proves our claim.

## Supporting information

**S1 Text. Supporting text.** sec-1 describes different competing methods and parameter settings, sec-2 describes description of copula in detail, sec-3 describes a short description of regularization techniques.
(PDF)

## Acknowledgments

We would like to thank Dr. Abhik Ghosh, Indian Statistical Institute, Kolkata for insightful discussions. SB acknowledges support from J.C. Bose Fellowship [SB/S1/JCB-033/2016 to S.B.] by the DST, Govt. of India.

## Author Contributions

**Conceptualization:** Sumanta Ray.

**Data curation:** Snehalika Lall, Sumanta Ray.

**Formal analysis:** Snehalika Lall, Sumanta Ray.

**Investigation:** Sumanta Ray, Sanghamitra Bandyopadhyay.

**Methodology:** Snehalika Lall, Sumanta Ray, Sanghamitra Bandyopadhyay.

**Project administration:** Sanghamitra Bandyopadhyay.

**Resources:** Snehalika Lall, Sumanta Ray.

**Software:** Snehalika Lall, Sumanta Ray.

**Supervision:** Sumanta Ray, Sanghamitra Bandyopadhyay.

**Validation:** Snehalika Lall, Sumanta Ray, Sanghamitra Bandyopadhyay.

**Visualization:** Snehalika Lall, Sumanta Ray.

**Writing – original draft:** Snehalika Lall, Sumanta Ray.

**Writing – review & editing:** Snehalika Lall, Sumanta Ray, Sanghamitra Bandyopadhyay.

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
