## [Decision Letter · Decision Letter 0]

23 Jul 2021

Dear Dr. Ray,

Thank you very much for submitting your manuscript "RgCop-A regularized copula based method for gene selection in single cell rna-seq data" for consideration at PLOS Computational Biology.

As with all papers reviewed by the journal, your manuscript was reviewed by members of the editorial board and by several independent reviewers. In light of the reviews (below this email), we would like to invite the resubmission of a significantly-revised version that takes into account the reviewers' comments.

We cannot make any decision about publication until we have seen the revised manuscript and your response to the reviewers' comments. Your revised manuscript is also likely to be sent to reviewers for further evaluation.

Sincerely,

Wei Li, Ph.D.

Guest Editor

PLOS Computational Biology

Ilya Ioshikhes

Deputy Editor

PLOS Computational Biology

Reviewer's Responses to Questions

**Comments to the Authors:**

Reviewer #1: Gene selection is an important step in single cell RNA-seq analysis. The authors proposed a novel regularized copula based method for feature selection. They applied their algorithm on synthetic data and four real scRNA-seq datasets. Their results show an improvement in the performance over the other state-of-the-art methods.

The main concern is whether RgCop outperform other methods. More evidences are needed to support the superiority of RgCop.

Major:

1. The synthetic data was generated using Gaussian distribution. However, scRNA-seq is affected by high dropout rate and the count data is usually very sparse which may follow negative binomial distribution. Now there are some methods for scRNA-seq synthetic data, such as splatter. The authors should try these simulation methods to assess RgCop’s performance.

2. In figure 3, RgCop was compared with four feature selection techniques. The authors select 10 sets of features ranging from 10 to 100. In most scRNA-seq study, the numbers of genes selected for downstream analysis are rarely under 100. Why do the authors use so few genes? When the selected feature numbers increase to 500, 1000 or 2000, does RgCop still have good performance over other methods? In addition, the stability should be evaluated with more genes.

3. Using genes selected by RgCop, can rare cell populations be identified?

4. In the Relevancy test, genes need to calculate Ccor score with class labels. What does “class label” mean? Is RgCop a supervised method to select genes? In my opinion, Kendall tau is used to measure ordinal association. Are class labels ordered?

5. In the case study of PBMC68k, the ARIs are quite low across all methods. This may be caused by the low gene numbers.

6. The authors split data in training and test set to prove RgCop’s predicting ability. This result is based on analysis within the same dataset. However, scRNA-seq data may be affected by batch effect. Can RgCop be trained in dataset and predicted accurately in another dataset? Is RgCop influenced by batch effect when it is utilized in a merged data? In addition, it seems RgCop can only predict cells from same tissues.

7. How fast is the algorithm? Have you compare the speed between RgCop and other algorithm as the cell number increases?

Reviewer #2: the review is uploaded as an attachment

Reviewer #3: In this study, the authors developed RgCop, a gene selection method based on regularized copula correlation for scRNA-seq data analysis. RgCop utilizes copula correlation to find the most informative genes from scRNA-seq datasets. In order to filter redundant genes, RgCop employs an l1 regularization term to punish the redundant coefficients of genes. The authors proved that RgCop is superior to other state-of-the-art methods in clustering cells, capturing the dependence between noisy data features, and annotating unknown cells. The main opinions and concerns we raised are listed below:

1. The author has provided demo data and RgCop software in Github, which allows users to use the demo data to test RgCop. But the author does not seem to mention how to use RgCop to analyze other scRNA-seq data. In order for the algorithm to be widely used, RgCop needs to allow users to enter their data and adjust parameters.

2. In lines 121 and 122, the authors mentioned “The features are selected by several supervised feature selection algorithm and the classification accuracy are compared with RgCop.” What algorithm was applied to select features?

3. Based on the Wilcoxon rank-sum test p-value of each cluster, the first five differentially expressed genes are selected as marker genes. These genes will be used to annotate the corresponding clusters. The annotation of each cluster can be directly affected by the marker gene. Therefore, how to select marker genes in scRNA-seq data analysis is a very difficult problem. Why did the authors decide to use the first five genes as marker genes?

4. In line 145, the authors selected 100 characteristic genes to cluster the cells. They then used the Adjusted Rand Index (ARI) to evaluate the clustering performance between the predicted cluster and the known group. Do the top 100 features make RgCop the best performance? Is this number of features consistent in the analysis of different scRNA-seq data? Figure 2A shows that the variation of the ARI value in the Muraro dataset is greater than the Yan and Pollen datasets during the process of changing the number of features from 10 to 100. The Muraro data set contains 2126 instances, which is more than the Yan and Pollen datasets. Does this mean that the size of the feature will significantly affect the performance of RgCop in the datasets with a large number of cells?

5. In the section “Classification performance on real dataset using supervised method”. Four classification methods were mentioned, but only three were presented. What method was used to measure the accuracy? What is the conclusion of this section?

6. In the abstract, the authors claim that “the differentially expressed (DE) genes identified from the clusters of scRNA-seq data are found to provide an accurate annotation of cells”. However, in the “Marker gene selection” section, the authors only described the differentially expressed genes of each dataset. Based on the result, how to conclude that RgCop can provide an accurate annotation of cells?

Minor points:

1. In the bottom panel of figure 1, the black text with a dark blue background is hard to read.

2. Line 110 on page 4: “doenstream” should be “downstream”.

**Have the authors made all data and (if applicable) computational code underlying the findings in their manuscript fully available?**

Reviewer #1: Yes

Reviewer #2: **No: **the code for de novo clustering the unlabeled dataset is missing

Reviewer #3: Yes

PLOS authors have the option to publish the peer review history of their article (what does this mean?). If published, this will include your full peer review and any attached files.

Reviewer #1: No

Reviewer #2: No

Reviewer #3: No
---

## [Decision Letter · Decision Letter 1]

19 Sep 2021

Dear Dr. Ray,

We are pleased to inform you that your manuscript 'RgCop-A regularized copula based method for gene selection in single cell rna-seq data' has been provisionally accepted for publication in PLOS Computational Biology.

Before your manuscript can be formally accepted you will need to complete some formatting changes, which you will receive in a follow up email. A member of our team will be in touch with a set of requests. In addition, please address the remaining minor issues raised by the reviewer.

Best regards,

Wei Li, Ph.D.

Guest Editor

PLOS Computational Biology

Ilya Ioshikhes

Deputy Editor

PLOS Computational Biology

Reviewer's Responses to Questions

**Comments to the Authors:**

Reviewer #1: Major comments:

1. It would be better to show results of “RgCop is robust for data with different batches”, e.g. UMAP plots of clustering result. Are CD141 cells and CD1C cells correctly discovered in cluster batch?

Minor comments:

1. Please check the title of table 5. The first letter of “overlaps” should be capitalized. “gees” may be “genes”?

2. It would be better to change tSNE to UMAP plot in Fig 5.

3. Please add a plot showing Leiden clusters in Fig 5B.

Reviewer #2: The authors have addressed all my concerns during revision and I don't have additional comments.

Reviewer #3: The ReCop method has been clarified in the revised manuscript. The superiority of ReCop over other state-of-the-art methods has been verified in datasets with different cell numbers. The author has addressed all my comments. I have no further comments.

**Have the authors made all data and (if applicable) computational code underlying the findings in their manuscript fully available?**

Reviewer #1: Yes

Reviewer #2: Yes

Reviewer #3: Yes

PLOS authors have the option to publish the peer review history of their article (what does this mean?). If published, this will include your full peer review and any attached files.

Reviewer #1: No

Reviewer #2: No

Reviewer #3: No

---

## [Editor Report · Acceptance letter]

6 Oct 2021

PCOMPBIOL-D-21-00847R1 

RgCop-A regularized copula based method for gene selection in single cell rna-seq data

Dear Dr Ray,

I am pleased to inform you that your manuscript has been formally accepted for publication in PLOS Computational Biology. Your manuscript is now with our production department and you will be notified of the publication date in due course.

With kind regards,

Olena Szabo
